# Rainmer: Learning Multi-view Representations for Comprehensive Image Deraining and Beyond

Wu Ran
School of Computer Science, Fudan University
Shanghai, China
wran21@m.fudan.edu.cn

Peirong Ma
School of Computer Science, Fudan University
Shanghai, China
prma20@fudan.edu.cn

Zhiquan He
School of Computer Science, Fudan University
Shanghai, China
22210240019@m.fudan.edu.cn

Hong Lu*
School of Computer Science, Fudan University
Shanghai, China
honglu@fudan.edu.cn

## Abstract

We address image deraining under complex backgrounds, diverse rain scenarios, and varying illumination conditions, representing a highly practical and challenging problem. Our approach utilizes synthetic, real-world, and nighttime datasets, wherein rich backgrounds, multiple degradation types, and diverse illumination conditions coexist. The primary challenge in training models on these datasets arises from the discrepancies among them, potentially leading to conflicts or competition during the training period. To address this issue, we first align the distribution of synthetic, real-world and nighttime datasets. Then we propose a novel contrastive learning strategy to extract multi-view (multiple) representations that effectively capture image details, degradations, and illuminations, thereby facilitating training across all datasets. Regarding multiple representations as profitable prompts for deraining, we devise a prompting strategy to integrate them into the decoding process. This contributes to a potent deraining model, dubbed Rainmer. Additionally, a spatial-channel interaction module is introduced to fully exploit cues when extracting multi-view representations. Extensive experiments on synthetic, real-world, and nighttime datasets demonstrate that Rainmer outperforms current representative methods. Moreover, Rainmer achieves superior performance on the All-in-One image restoration dataset, underscoring its effectiveness. Furthermore, quantitative results reveal that Rainmer significantly improves object detection performance on both daytime and nighttime rainy datasets. These observations substantiate the potential of Rainmer for practical applications.

## CCS Concepts

• **Computing methodologies → Reconstruction**.

*Corresponding author.

## Keywords

Image deraining, multi-view representations, prompting deraining

**ACM Reference Format:**
Wu Ran, Peirong Ma, Zhiquan He, and Hong Lu. 2024. Rainmer: Learning Multi-view Representations for Comprehensive Image Deraining and Beyond. In *Proceedings of the 32nd ACM International Conference on Multimedia (MM '24), October 28-November 1, 2024, Melbourne, VIC, Australia.* ACM, New York, NY, USA, 10 pages. https://doi.org/10.1145/3664647.3681342

## 1 Introduction

Rain occurs approximately eight times more frequently than fog and snow according to the statistics of the latest large autonomous driving dataset [1]. As the most common adverse weather condition, rain significantly impairs outdoor computer vision applications. Recently, the sixth UG$^2$+ Prize Challenge highlighted the importance and urgency of real-world image deraining tasks.

Deep learning-based methods [4, 16, 30, 50–52, 54, 57, 58, 63] have made significant progress in image deraining in decades. However, the majority of these algorithms are primarily trained on synthetic datasets [4, 16, 30, 51, 52, 54, 57], which limits their effectiveness in real-world and nighttime scenarios. Consequently, a fundamental question arises: *What factors contribute to the challenges in establishing a comprehensive image deraining model?*

Constructing paired real-world and nighttime datasets has long been challenging in the image de-raining literature. Because capturing rainy images and clean images simultaneously while maintaining pixel-wise consistency is usually rendered impossible. Therefore, the majority of methods rely on synthetic datasets [8, 20, 51, 63] to develop powerful deraining models, which often struggle in real-world rainy environments and nighttime scenarios. Recently, Wang *et al.* [45] proposes to generate pseudo clean backgrounds from real rainy videos to create paired datasets. Subsequently, Ba *et al.* [2] and Zhang *et al.* [62] introduce a time multiplexing technique to collect paired rainy and clean images from online video streams at different timestamps. Though effective, these methods mainly collect images in static scenes with little motion movement and cannot be adapted to heavy rainy scenes as well as low-light conditions, restricting the richness of backgrounds and illuminations. Most recently, Zhang *et al.* [60] create a paired nighttime rainy dataset leveraging the weather simulation technique in the GTAV game.

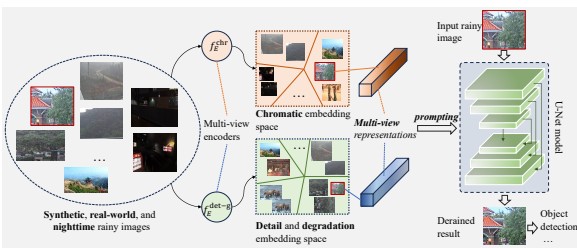

**Figure 1: Motivation of the proposed multi-view representation learning-based image deraining.**

Hence, a more feasible approach to developing comprehensive deraining models is to combine the advantages of synthetic, real-world, and nighttime datasets. The benefits are three-fold: 1) **Rich background scenes** from synthetic datasets. Synthetic datasets [8, 20, 52] usually utilize high-quality images from BSD [29] and UCID [38] to synthesize rainy images; 2) **Rich degradations** from synthetic and real-world datasets. Synthetic approaches can control the rain by tuning hyper-parameters, while the real-world dataset contains complex rain effects, *e.g.*, rain streaks, rain veiling effect, blur, and color distortions; 3) **Diverse ambient lighting conditions** from nighttime datasets. Rain in the daytime is often monochromatic [59, 65] while being colored by ambient lights in nighttime scenarios [60]. Current approaches [16, 30, 57, 58] attempt to acquire decent deraining ability by directly training models on mixed synthetic datasets. However, this approach suffers from larger dataset discrepancies arising from various degradations and significant illumination contrasts. There are also All-in-One methods [19, 21, 67] developed to address multiple weather conditions, but they pay little attention to the diversity and complexity of rain itself. Additionally, we observe significant differences in the rain density distribution between daytime and nighttime datasets, which may impede the learning process.

To address the aforementioned issues, we first align the rain density distributions among synthetic, real-world, and nighttime datasets. Typically, we find that a large number of black blocks in rainy night images are meaningless for training and should be removed. Then, we aim to extract multi-view (multiple) representations from rainy images that characterize image details, degradations, and illuminations, as illustrated in Fig. 1. Existing representation learning approaches focus on learning degradation-related representation [19, 44], discriminating between rain and backgrounds [5, 56] and extracting joint rain-/detail-aware representation [35], which are not capable of simultaneously perceiving image details, degradations, and illuminations. Inspired by [35], we utilize detail-/degradation-aware representation to capture image backgrounds, rain streaks, and blur effects. This joint representation facilitates learning overall rain densities. Moreover, a chromatic representation is explored to capture color distortions, rain veiling effects, and illuminations. These two kinds of representations constitute multi-view representations, which are efficiently learned with the proposed contrastive learning approach in an unsupervised manner. These representations are expected to model discrepancies among datasets well and promote dataset collaboration. Beyond, they serve as valuable prompts to guide the deraining process as

shown in Fig. 1. Hence, we further devise a prompting strategy that contributes to Rainmer's design. Unlike current prompting strategies [33, 39] which employ a fixed number of prompts, the proposed Rainmer directly extracts prompts from input images, functioning effectively at larger dataset scales. Additionally, Rainmer includes a Spatial-channel Interaction Module (SCIM), which facilitates the full exploitation of channel information when extracting multi-view (multiple) representations. In this paper, we make the following contributions:

1) We propose to learn a comprehensive image deraining model leveraging the combinations of synthetic, real-world, and nighttime datasets. A rain density distribution alignment strategy is introduced to mitigate gaps among these datasets.

2) To effectively learn from synthetic, real-world, and nighttime datasets, we propose a multi-view representation learning method aimed at capturing backgrounds, degradations, and illuminations. On top of these representations, we develop Rainmer, which utilizes multiple representations to prompt image deraining. Additionally, a SCIM module is incorporated into Rainmer.

3) We conduct extensive experiments on synthetic, real-world, and nighttime datasets, where the proposed Rainmer outperforms current state-of-the-art. Remarkable performance on the All-in-One image restoration dataset further emphasizes the superiority of the proposed approach. Moreover, experimental results on downstream object detection indicate a significant improvement in mean Average Precision (mAP) on both daytime and nighttime datasets.

## 2 Related Work

**Single Image Deraining** focuses on the removal of rain effects to restore clean backgrounds for outdoor computer vision applications. Prior-based deraining methods, utilizing techniques such as Gaussian Mixture Models [23] and dictionary learning [10, 27], achieve this goal through iterative optimization but often incur significant computational burdens and struggle with generalization. With the advent of deep learning, learning-based approaches [8, 30, 35, 43, 52, 54, 58] have emerged, dramatically enhancing deraining ability over the past decades. In the realm of learning-based deraining, multi-scale design [9, 42, 53], attention mechanisms [4, 13, 41, 45], recurrent units [36, 37, 54], and multi-stage processing [52, 58] have been extensively explored to adress complex and accumulated rain streaks. There are contemporary semi-supervised and unsupervised methods [5, 14, 47, 55, 66] that attempt to incorporate unlabeled real-world data for training, often yielding undesired performance. Current outstanding deraining methods [6, 33, 50] primarily focus on synthetic datasets, with limited attention given to real-world and nighttime environments. Researchers have begun to develop real-world rainy datasets [2, 62], as well as nighttime dataset [60, 61]. Albeit successful, comprehensive image deraining with complex backgrounds, diverse rain degradations, and varying illumination conditions remains a largely unexplored and urgent problem.

**Representation Learning-based Image Restoration** aims to produce high-quality results by leveraging interactions between intermediate features and abstract representations. Wei *et al.* [47] represent synthetic and real rain using Gaussian Mixture Models, where Kullback-Leiber divergence [7] is imposed to transfer knowledge from synthetic images to unlabeled real rainy images.

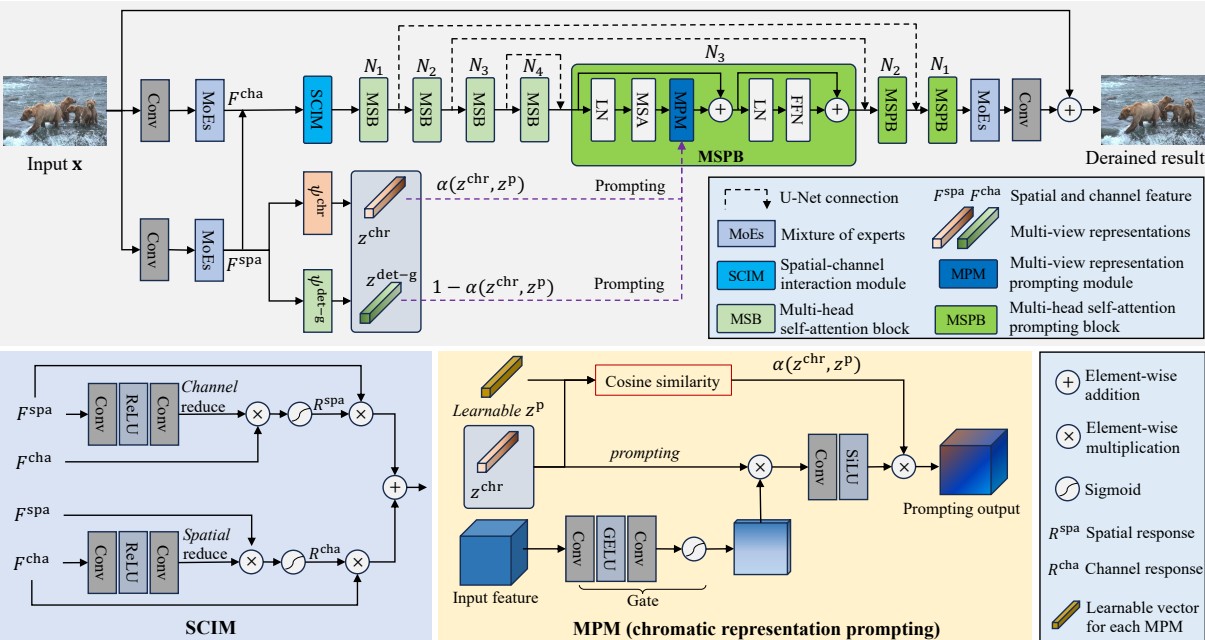

**Figure 2: The architecture of the proposed Rainmer. Rainmer initially extracts multi-view representations $z^{\text{chr}}$ and $z^{\text{det-g}}$ from input image using mixture of experts (MoEs). Then, Rainmer employs a U-Net-based architecture to remove rain effects, where we develop MPM to prompt the decoding process utilizing multiple representations. Additionally, we propose SCIM to maximize spatial-channel interaction when extracting representations.**

Huang *et al.* [14] introduce a soft-updating strategy to facilitate interaction among representations of synthetic and real rainy images. Recently, contrastive learning [12] has been widely adopted in image restoration [19, 44, 49, 56]. Typically, Wang *et al.* [44] extract degradation representations with contrastive learning to guide image super-resolution. Li *et al.* [19] employ degradation representations to generate parameters of deformable convolution layers, thereby enhancing All-in-One image restoration. In contrast, Transweather [39] incorporates learnable weather-type queries to facilitate feature interaction. Following [39], Potlapalli *et al.* [33] introduce a fixed number of learnable prompts to guide the image restoration process. Leveraging a pre-trained model to extract degradation representation, Wang *et al.* [40] devise a prompting image restorer. Though effective, these methods mainly focus on utilizing degradation-aware representations, neglecting image details and illumination. Most recently, Ran *et al.* [35] propose to learn joint rain-/detail-aware representations to remove complex rain effects, while overlooking illumination differences.

## 3 Comprehensive Image Deraining

In this paper, we aim to address image deraining leveraging synthetic, real-world, and nighttime datasets. Denote $\mathcal{D}^{\text{syn}}$, $\mathcal{D}^{\text{real}}$, and $\mathcal{D}^{\text{night}}$ as the synthetic, real-world and nighttime datasets, respectively. And let $\bar{\mathcal{D}} = \{\mathbf{x}_i, \mathbf{y}_i\}_{i=1}^{N}$ be their union, where $\mathbf{x}_i$ and $\mathbf{y}_i$ represent $i$-th rainy and clean images, and $N$ is the total number of samples. To handle discrepancies among these datasets, we propose to learn multiple representations that effectively perceive image details, degradations, and illuminations.

Specifically, given rainy image $\mathbf{x}$, we employ a chromatic encoder $f_E^{\text{chr}}$ and a detail-/degradation-aware encoder $f_E^{\text{det-g}}$ (see Fig. 1) to extract corresponding representations following:

$$z^{\text{chr}} = f_E^{\text{chr}}(\mathbf{x}), \quad z^{\text{det-g}} = f_E^{\text{det-g}}(\mathbf{x}), \tag{1}$$

where $z^{\text{chr}}$ and $z^{\text{det-g}}$ are $d$-dimensional spherical vectors, indicating multi-view (multiple) representations. We introduce a novel contrastive learning strategy to learn $f_E^{\text{chr}}$ and $f_E^{\text{det-g}}$, with training objective formulated as:

$$\mathcal{L}_{\text{contra}} = \mathcal{L}_{\text{contra}}^{\text{chr}} + \mathcal{L}_{\text{contra}}^{\text{det-g}}, \tag{2}$$

where $\mathcal{L}_{\text{contra}}^{\text{chr}}$ and $\mathcal{L}_{\text{contra}}^{\text{det-g}}$ denote contrastive learning losses that supervise the training of $f_E^{\text{chr}}$ and $f_E^{\text{det-g}}$. Fig. 2 presents the details of encoders, which share a mixture of experts (MoEs) and possess specific feature projectors denoted as $\psi^{\text{chr}}$ and $\psi^{\text{det-g}}$. The representation learning procedure will be elaborated in Section 3.2.

Aside from effectively perceiving details, degradations, and illuminations, the multi-view representations could be exploited to assist the deraining procedure. To this end, we further devise Rainmer, which contains Multi-view representations Prompting Modules (MPM) to prompt the decoding process, as depicted in Fig. 2. In addition to the vector-level prompting in MPM, we introduce a Spatial-channel Interaction Module (SCIM) to exploit rich spatial-channel interaction when extracting representations. Details of Rainmer will be discussed in Section 3.3. Generally, this prompting deraining process can be formulated as:

$$\mathbf{y}^{\text{p}} = f(\mathbf{x}, z^{\text{chr}}, z^{\text{det-g}}), \tag{3}$$

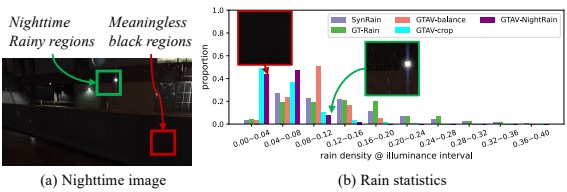

(a) Nighttime image · (b) Rain statistics

**Figure 3: Rain density @ illuminance statistical analysis across all datasets.**

where $\mathbf{y}^p$ is the restored image, and $f$ represents the deraining process of Rainmer as shown in Fig. 2. The prediction of restored image $\mathbf{y}^p$ can be supervised with a reconstruction loss:

$$\mathcal{L}_{\text{recon}} = \mathbb{E}_{(\mathbf{x},\mathbf{y})} \left[ \sqrt{||\mathbf{y} - \mathbf{y}^p||_F^2 + \epsilon^2} + 0.1(1 - \text{SSIM}(\mathbf{y}, \mathbf{y}^p)) \right], \quad (4)$$

where we employ the combination of Charbonnier loss [17] (first term) and SSIM [46] loss (second term), $\epsilon = 10^{-3}$. In summary, We train Rainmer end-to-end using the combination of $\mathcal{L}_{\text{recon}}$ and $\mathcal{L}_{\text{contra}}$:

$$\mathcal{L}_{\text{tot}} = \mathcal{L}_{\text{recon}} + \lambda \mathcal{L}_{\text{contra}}, \quad (5)$$

where $\lambda$ is a hyper-parameter.

## 3.1 Dataset Distribution Alignment

The ambient lighting conditions may induce a large gap between daytime and nighttime datasets, which is harmful to training. We investigate the rain density distribution of synthetic, real-world, and nighttime datasets. Typically, we choose Rain13K [16] (*synthetic*, noted as SynRain), GT-Rain [2] (*real-world*), and GTAV-NightRain [60] (*nighttime*) datasets for analysis. Note that rainy images captured at night usually have much lower pixel intensities compared to daytime images, which influences rain density calculation. Hence we introduce a rain density @ illuminance metric:

$$\rho = \frac{\max_c \sum_{h,w} |\mathbf{x}_{h,w,c} - \mathbf{y}_{h,w,c}|}{\max_c \sum_{h,w} \mathbf{x}_{h,w,c} + \max_c \sum_{h,w} \mathbf{y}_{h,w,c}}, \quad (6)$$

where $\mathbf{x}$ and $\mathbf{y}$ are paired rainy and clean images. $h, w$ represents spatial coordinates, and $c$ means color channel index. The metric $\rho$ in Eq. (6) describes *the extent to which the image is corrupted by rain when compared to the overall pixel intensity of the background and rainy image*. Fig. 3 (b) presents the statistics of $\rho$ among all datasets. It can be seen that SynRain and GT-Rain share similar statistics of $\rho$, while presenting a large difference against GTAV-NightRain. Specifically, images with $\rho < 0.08$ in GTAV-NightRain account for about 90% proportion. In fact, we observe that this is due to plenties of black regions in nighttime images with $\rho \approx 0$ (see Fig. 3 (a)), which contains almost no information and is meaningless. To address this problem, we first crop all images in GTAV-NightRain into non-overlap $512 \times 512$ images, resulting in GTAV-crop. Then we remove a proportion of images with $\rho < 0.08$ in GTAV-crop by matching the average statistics of $\rho$ in the intervals $[0.00, 0.04]$ and $[0.04, 0.08]$ of SynRain and GT-Rain. Finally, we obtain the GTAV-balance dataset, which presents similar statistics to SynRain and GT-Rain as illustrated in Fig. 3 (b).

## 3.2 Multi-view Representation Learning

Contrastive learning [12] has been proven an elegant and successful approach to learning image representation in an unsupervised manner [19, 35, 44]. In this section, we depict the proposed contrastive learning method to learn multi-view representations in detail. To extract representation as shown in Fig. 2 from $\mathbf{x}$, we first utilize a mixture of experts to obtain low-level spatial features:

$$F^{\text{spa}} = f_{\text{MoE}} \circ f_{\text{conv}}(\mathbf{x}), \quad (7)$$

where $f_{\text{MoE}}$ indicates the mixture of experts, and $f_{\text{conv}}$ is a convolutional layer to expand channels. In practice, we implement $f_{\text{MoE}}$ with eight experts following [4]. $F^{\text{spa}}$ is the low-level feature with rich spatial information to perceive image details, degradations, and illuminations. Utilizing chromatic projector $\psi^{\text{chr}}$ and detail-/degradation-aware projector $\psi^{\text{det-g}}$ containing three convolutional layers, we extract chromatic information and detail/degradation information from $F^{\text{spa}}$ by:

$$F^{\text{chr}} = \psi^{\text{chr}}(F^{\text{spa}}), \ F^{\text{det-g}} = \psi^{\text{det-g}}(F^{\text{spa}}), \quad (8)$$

where $F^{\text{chr}}$ and $F^{\text{det-g}}$ denote corresponding deep features, respectively. Details of $\psi^{\text{chr}}$ and $\psi^{\text{det-g}}$ are provided in the supplementary materials. Then we can obtain image chromatic representation $z^{\text{chr}}$ and degradation representation $z^{\text{deg}}$ following:

$$z^{\text{chr}} = \frac{\text{GAP}(F^{\text{chr}})}{||\text{GAP}(F^{\text{chr}})||_2}, \ z^{\text{deg}} = \frac{\text{GAP}(F^{\text{det-g}})}{||\text{GAP}(F^{\text{det-g}})||_2}, \quad (9)$$

where $\text{GAP}(\cdot)$ means global average pooling operation. Note that GAP will destroy the spatial information in feature, hence we obtain detail representation $Z^{\text{det}}$ while preserving spatial information via:

$$Z^{\text{det}} = \frac{\text{Pool}(F^{\text{det-g}})}{||\text{Pool}(F^{\text{det-g}})||_2}, \quad (10)$$

where $\text{Pool}(\cdot)$ is a pooling operation with $8 \times 8$ kernel and stride 8.
**Detail-/Degradation-aware Representation Learning**. Ran *et al.* [35] have developed an efficient approach to extract joint rain-/detail-aware representations. The underlying philosophy is that by pushing rainy image $\mathbf{x}$ apart from negatives with blurred backgrounds and negatives with most dissimilar rain effects simultaneously, the model can in turn learn rain-/detail-aware representations. Inspired by this, we construct $N_b$ detail-aware samples by employing Gaussian blur on clean image $\mathbf{y}$ following [35], denoted as $\{\mathbf{y}_j^b\}_{j=1}^{N_b}$. To construct negatives for learning degradation, we maintain a rain layer archive $\mathcal{A}_r$ similar to [35]. Different from [35], given rainy image $\mathbf{x}$, we retrieve *most dissimilar* rain layers from $\mathcal{A}_r$ to construct negatives by:

$$\{\mathbf{r}_j\}_{j=1}^{N_r} = \underset{A \subset \mathcal{A}_r, |A|=N_r}{\arg\max} \sum_{\mathbf{r}' \in A} ||\text{svd}(\mathbf{r}) - \text{svd}(\mathbf{r}')||_1, \ \mathbf{r} = \mathbf{x} - \mathbf{y}, \quad (11)$$

where $N_r$ means the number of degradation-aware negatives, $\mathbf{r}$ and $\mathbf{r}'$ denote the rain layer of $\mathbf{x}$ and rain layer in archive, respectively. The $\text{svd}(\cdot)$ calculates the singular values in descending order, which is robust to translation and rotation. Utilizing detail-aware negatives $\{\mathbf{y}_j^b\}_{j=1}^{N_b}$ and degradation-aware negatives simulated by $\mathbf{y} + \mathbf{r}_j$, we calculate corresponding negative logits for contrastive learning by:

$$s_j^{\text{det}} = Z^{\text{det}} \cdot Z_j^{\text{det}}, \ s_j^{\text{deg}} = z^{\text{deg}} \cdot z_j^{\text{deg}}, \quad (12)$$

where $Z_j^{\text{det}}$ and $z_j^{\text{deg}}$ are calculated using Eq. (10) and Eq. (9), corresponding to $\mathbf{y}_j^b$ and $\mathbf{y} + \mathbf{r}_j$, respectively. By employing data augmentation to $\mathbf{x}$, we can calculate positaive logits $s^{\text{det+}}$ and $s^{\text{deg+}}$ similar to Eq. (12). Hence, the contrastive learning loss $\mathcal{L}_{\text{contra}}^{\text{det-g}}$ in Eq. (2) is formulated as:

$$\mathcal{L}_{\text{contra}}^{\text{det-g}} = -\log\left(\frac{e^{s^{\text{det+}}+s^{\text{deg+}}}}{e^{s^{\text{det+}}+s^{\text{deg+}}} + \sum_{j=1}^{N_b} e^{s_j^{\text{det}}} + \sum_{j=1}^{N_r} e^{s_j^{\text{deg}}}}\right). \quad (13)$$

When prompting restoration, $Z^{\text{det}}$ and $z^{\text{deg}}$ are combined to $z^{\text{det-g}}$ since they both come from $F^{\text{det-g}}$.

**Chromatic Representation Learning**. To perceive color distortions and illumination in rainy images, we compute a chromatic vector $\mathbf{u} \in \mathbb{R}^3$ for image $\mathbf{x}$ via:

$$\mathbf{u}_c = \frac{1}{HW} \sum_{h,w} \mathbf{x}_{h,w,c}, \ c \in \{R, G, B\}, \quad (14)$$

where $H$ and $W$ denote the spatial resolution, and $\mathbf{u}$ characterizes both the color property and overall illuminance. We can re-render the chromatic property of $\mathbf{x}$ with another $\mathbf{u}'$ by:

$$\tilde{\mathbf{x}} = (\mathbf{u}'/\sum_c \mathbf{u}_c) \odot \mathbf{x}, \quad (15)$$

where $\odot$ means element-wise multiplication, and $\tilde{\mathbf{x}}$ is the re-rendering result. Eq. (15) provides a way to construct chromatic-aware negatives. To this end, we maintain a chromatic vector archive $\mathcal{A}_c$ and retrieve $N_c$ *most dissimilar* vectors $\{\mathbf{u}_j\}_{j=1}^{N_c}$ from it given rainy image $\mathbf{x}$. The $\{\mathbf{u}_j\}_{j=1}^{N_c}$ is then employed to simulate chromatic-aware negatives $\{\tilde{\mathbf{x}}_j\}_{j=1}^{N_c}$ using Eq. (15). Similar to Eqs. (12) and (13), we derive chromatic contrastive loss $\mathcal{L}_{\text{contra}}^{\text{chr}}$ in Eq. (2) below:

$$\mathcal{L}_{\text{contra}}^{\text{chr}} = -\log\left(\frac{e^{z^{\text{chr}} \cdot z^{\text{chr+}}}}{e^{z^{\text{chr}} \cdot z^{\text{chr+}}} + \sum_{j=1}^{N_c} e^{z^{\text{chr}} \cdot z_j^{\text{chr}}}}\right), \quad (16)$$

where $z^{\text{chr}}$, $z^{\text{chr+}}$, and $z_j^{\text{chr}}$ are chromatic representations for rainy image $\mathbf{x}$, clean image $\mathbf{y}$, and negatives $\tilde{\mathbf{x}}_j$, respectively. By pulling chromatic representations of $\mathbf{x}$ and $\mathbf{y}$ together, the proposed method could implicitly address color distortions raised by rain effects.

## 3.3 Rainmer: Prompting Deraining

As shown in Fig. 2, Rainmer first employs MoEs to extract low-level features $F^{\text{cha}}$, and then processes features through a 4-level encoder-decoder architecture. In the end, the features from the decoder undergo another MoEs to obtain final restored result. At the $l$-th level, the encoder comprises $N_l$ Multi-head Self-attention Blocks (MSB) as shown in Fig. 2. Each MSB is a LayerNorm (LN)-Multi-head Self-attention (MSA)-LayerNorm-Feedforward Network (FFN) architecture (see Fig. 2). The $l$-th decoder contains $N_l$ Multi-head Self-attention Prompting Blocks (MSPB), with a Multi-view Prompting Module (MPM) inserted behind MSA of the MSB as illustrated in Fig. 2. The MPM utilizes multi-view representations to prompt the decoding process, thereby producing high-quality results. Additionally, we introduce a Spatial-channel Interaction Module (SCIM) to enhance the low-level features $F^{\text{cha}}$. In practice,

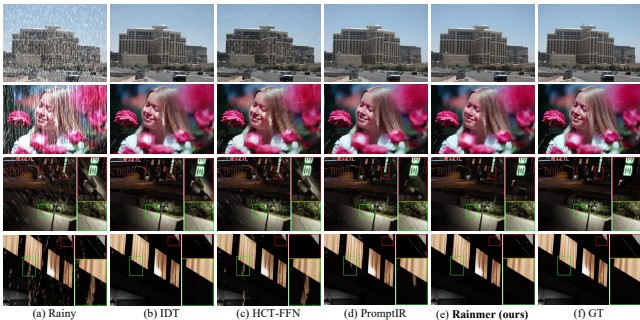

|  |  |  |  |  |  |
|---|---|---|---|---|---|
| (a) Rainy | (b) IDT | (c) HCT-FFN | (d) PromptIR | (e) **Rainmer (ours)** | (f) GT |

**Figure 4: Visual comparison on image deraining. The results on the first and second rows are from synthetic datasets Test100 and Rain100H, respectively. The results on the last two rows are from GTAV-balance.**

we implement the MoEs, MSA, and FFN following [4], where sparse attention and mixed-scale feedforward network are employed.

**Multi-view Prompting Module (MPM)**. Given an input feature $F^{\text{in}}$, multi-view representations $z^{\text{chr}}$ and $z^{\text{det-g}}$, MPM first computes spatial response maps $R^{\text{chr}}$ and $R^{\text{det-g}}$ corresponding to $z^{\text{chr}}$ and $z^{\text{det-g}}$ via:

$$R^t = \mathcal{G}^t(F^{\text{in}}), \ t \in \{\text{chr}, \text{det-g}\}, \quad (17)$$

where $\mathcal{G}^t, t \in \{\text{chr}, \text{det-g}\}$ represents a Conv-GELU-Conv-Sigmoid gate. Utilizing the response map, MPM integrates multi-view representations by:

$$F_p^t = \mathcal{H}^t(R^t \odot z^t), \ t \in \{\text{chr}, \text{det-g}\}, \quad (18)$$

where $\mathcal{H}^t, t \in \{\text{chr}, \text{det-g}\}$ denotes a Conv-SiLU transformation. To merge $F_p^t, t \in \{\text{chr}, \text{det-g}\}$, MPM employs a learnable vector $z^{\text{p}}$ to calculate the response of chromatic representation by $\alpha(z^{\text{chr}}, z^{\text{p}}) = (1 + z^{\text{chr}} \cdot z^{\text{p}})/2$. Finally, MPM produces weighted output:

$$F^{\text{out}} = \alpha(z^{\text{chr}}, z^{\text{p}})F_p^{\text{chr}} + (1 - \alpha(z^{\text{chr}}, z^{\text{p}}))F_p^{\text{det-g}}. \quad (19)$$

**Spatial-channel Interaction Module (SCIM)**. As shown in Eqs. (8) to (10), all multi-view representations are from $F^{\text{spa}}$, which contains rich information. Hence, we propose the SCIM to explore the mutual spatial-channel interaction between $F^{\text{spa}}$ and low-level features $F^{\text{cha}}$ from MoEs. SCIM subsequently outputs an enhanced feature for removing various rain effects as well as restoring details. Fig. 2 presents the details of SCIM.

## 4 Experiments

In this section, we conduct extensive experiments to assess the efficacy of our approach. Specifically, negative numbers $N_b$, $N_r$, and $N_c$ in Section 3.2 are set to 4 following [35, 48]. The capacities of archives $\mathcal{A}_r$ and $\mathcal{A}_c$ are 256, which are dynamically updated with the current data batch in a queue-like manner. The dimensions of all representations are 128 following [12]. In Rainmer, the numbers of MSB (MSPB) in each encoder (decoder) level are [4, 6, 6, 8]. Rainmer comprises about 36M parameters, with an increase of 2M parameters compared to [4]. We implement Rainmer using the PyTorch [32] framework. During training, the hyper-parameter $\lambda$ in Eq. (5) is set to 0.1. The batch size is 8 with $128 \times 128$ patch size,

**Table 1: Quantitative comparison on synthetic, real-world, and nighttime datasets in terms of PSNR and SSIM metrics. The best and second results are bolded and underlined, respectively.**

| Method | Rain100L | | Rain100H | | Test100 | | Test1200 | | Test2800 | | GT-Rain | | GTAV-balance | | Avg. | |
|---|---|---|---|---|---|---|---|---|---|---|---|---|---|---|---|---|
| | PSNR↑ | SSIM↑ | PSNR↑ | SSIM↑ | PSNR↑ | SSIM↑ | PSNR↑ | SSIM↑ | PSNR↑ | SSIM↑ | PSNR↑ | SSIM↑ | PSNR↑ | SSIM↑ | PSNR↑ | SSIM↑ |
| Rainy | 26.90 | 0.8384 | 13.55 | 0.3786 | 22.55 | 0.7035 | 23.64 | 0.7794 | 24.36 | 0.8108 | 21.20 | 0.6325 | 26.39 | 0.8015 | 22.66 | 0.7064 |
| PReNet [37] (CVPR'19) | 29.22 | 0.9212 | 25.45 | 0.8355 | 23.71 | 0.8523 | 29.62 | 0.9193 | 30.43 | 0.9317 | 22.32 | 0.6608 | 30.85 | 0.9310 | 27.37 | 0.8645 |
| BRN [36] (TIP'20) | 29.88 | 0.9251 | 26.53 | 0.8495 | 24.11 | 0.8584 | 29.92 | 0.9237 | 30.97 | 0.9376 | 22.10 | 0.6621 | 31.44 | 0.9376 | 27.85 | 0.8706 |
| RCDNet [43] (CVPR'20) | 29.26 | 0.9110 | 26.66 | 0.8244 | 24.50 | 0.8492 | 29.76 | 0.9167 | 30.60 | 0.9296 | 22.47 | 0.6599 | 31.12 | 0.9243 | 27.77 | 0.8593 |
| EfDerain [11] (AAAI'21) | 30.66 | 0.9253 | 26.95 | 0.8411 | 25.30 | 0.8739 | 30.95 | 0.9239 | 30.95 | 0.9328 | 22.91 | **0.6779** | 32.52 | 0.9433 | 28.61 | 0.8740 |
| IDT [50] (TPAMI'22) | 34.67 | 0.9619 | 27.93 | 0.8754 | 27.51 | 0.9108 | 30.37 | **0.9383** | 32.26 | 0.9505 | 22.48 | 0.6604 | 34.56 | 0.9582 | 29.97 | 0.8936 |
| AirNet [19] (CVPR'22) | 28.67 | 0.8853 | 26.17 | 0.7964 | 24.26 | 0.8388 | 31.02 | 0.9211 | 31.12 | 0.9332 | 22.57 | 0.6505 | 31.07 | 0.9296 | 27.84 | 0.8507 |
| unsup. NLCL [56] (CVPR'22) | 20.42 | 0.8287 | 17.92 | 0.5001 | 21.38 | 0.7663 | 22.90 | 0.8183 | 23.32 | 0.8508 | 21.93 | 0.6237 | 27.23 | 0.8817 | 22.16 | 0.7528 |
| HCT-FFN [6] (AAAI'23) | 29.98 | 0.9286 | 26.57 | 0.8473 | 24.75 | 0.8733 | 30.83 | 0.9342 | 30.95 | 0.9409 | 23.12 | 0.6633 | 31.32 | 0.9305 | 28.22 | 0.8740 |
| DRSformer [4] (CVPR'23) | 34.75 | 0.9538 | 28.83 | 0.8554 | 27.78 | 0.8907 | 31.96 | 0.9375 | 33.01 | 0.9515 | 23.56 | 0.6633 | 34.64 | 0.9592 | 30.65 | 0.8873 |
| PromptIR [33] (NIPS'23) | 35.30 | 0.9630 | 28.86 | 0.8644 | 28.70 | 0.8962 | 29.04 | 0.9123 | 32.14 | 0.9479 | 23.42 | 0.6692 | 34.78 | 0.9600 | 30.32 | 0.8876 |
| CoIC [35] (ICLR'24) | 34.86 | 0.9524 | 29.00 | 0.8589 | 28.69 | 0.9000 | **31.99** | 0.9374 | 33.03 | 0.9522 | **23.84** | 0.6664 | 34.50 | 0.9595 | 30.84 | 0.8895 |
| Rainmer (**ours**) | **36.42** | **0.9669** | **29.38** | **0.8775** | **29.86** | **0.9171** | 31.98 | 0.9377 | **33.22** | **0.9543** | 23.17 | 0.6594 | **35.42** | **0.9631** | **31.35** | **0.8966** |

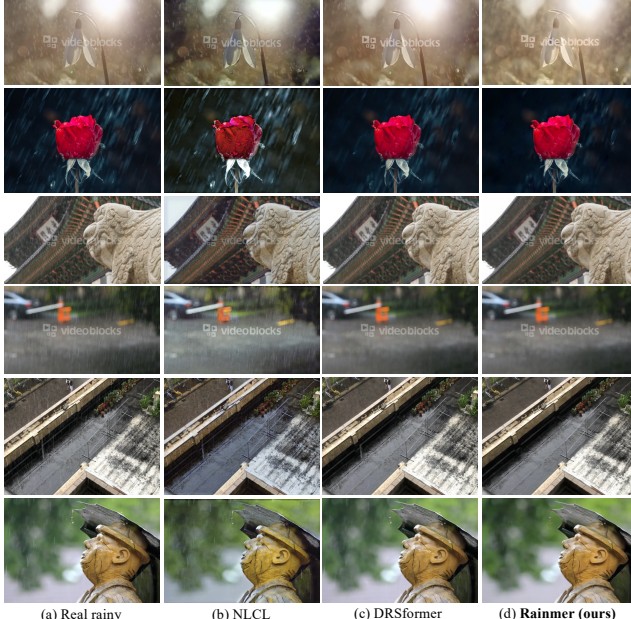

(a) Real rainy     (b) NLCL     (c) DRSformer     (d) **Rainmer (ours)**

**Figure 5: Visual comparison on real-world rainy images collected on the Internet.**

and the gradient accumulation technique is employed. All experiments are conducted on an NVIDIA Tesla V100 GPU. We adopt the AdamW optimizer [26] to train Rainmer for 300K iterations. The learning rate starts at $3e^{-4}$ for the initial 92K iterations and then decreases to $1e^{-6}$ for the remaining 208K iterations using the cosine annealing scheme [25].

### 4.1 Datasets and Evaluation Metrics

**Datasets**. We select Rain13K [16], GT-Rain [2], and the GTAV-balance dataset in Section 3.1 for benchmarking image deraining methods. Specifically, the *synthetic* Rain13K comprises 13,712 image pairs for training, along with 5 testing sets: Rain100L [52], Rain100H [52], Test100 [64], Test1200 [63], and Test2800 [8]. The

challenging *real-world* GT-Rain comprises 26,124 pairs for training and 2100 for testing. The *nighttime* GTAV-balance, inherited from GTAV-NightRain [60], contains 12,321 pairs for training and 3587 for evaluation, all with a resolution of $512 \times 512$. Additionally, we conduct experiments on an All-in-One image restoration dataset AllWeather [19], which includes 18,069 training pairs and three testing sets: Outdoor-Rain [20], RainDrop [34], and Snow100K-L [24]. For the comparison of object detection in night rainy scenes, we utilize the set3 testing set of GTAV-NightRain, containing 1860 samples with resolutions of $1920 \times 1080$.

**Metrics**. For quantitative comparison of image deraining and All-in-One image restoration, we adopt widely used PSNR [15] and SSIM [46] metrics following [31, 57]. Additionally, we utilize the mean average precision [3] (mAP) metric to evaluate object detection performance across different intersection over union (IoU) thresholds.

### 4.2 Comparison on Image Deraining

We compare the proposed Rainmer with eleven representative methods, including six CNN-based methods (PReNet [37], BRN [36], RCDNet [43], EfDerain [11], AirNet [19], and unsupervised NLCL [56]), four recent Transformer methods (IDT [50], DRSformer [4], PromptIR [33], and CoIC [35] with DRSformer backbone), and the hybrid CNN-Transformer method HCT-FFN [6]. Note that AirNet and CoIC are contrastive learning-based methods, while PromptIR is prompting-based. To ensure fair comparisons, we re-train all selected methods on mixed Rain13K, GT-Rain, and GTAV-balance datasets following their official instructions. Quantitative results are tabulated in Table 1. Rainmer has outperformed all selected methods with the highest average PSNR/SSIM metric across all datasets, substantiating the superiority of the proposed method. Specifically, Rainmer has brought PSNR improvements of 1.12dB, 0.38dB, 1.16dB, 0.19dB, and 0.64dB over the previous best results on four synthetic datasets (Rain100L, Rain100H, Test100, and Test2800) and the GTAV-balance dataset, respectively. Without chromatic representation, both AirNet and CoIC fail to obtain outstanding results on GTAV-balance. Surprisingly, NLCL obtains the worst performance, demonstrating huge challenge of unsupervised deraining with large rain discrepancies. However, Rainmer fails to

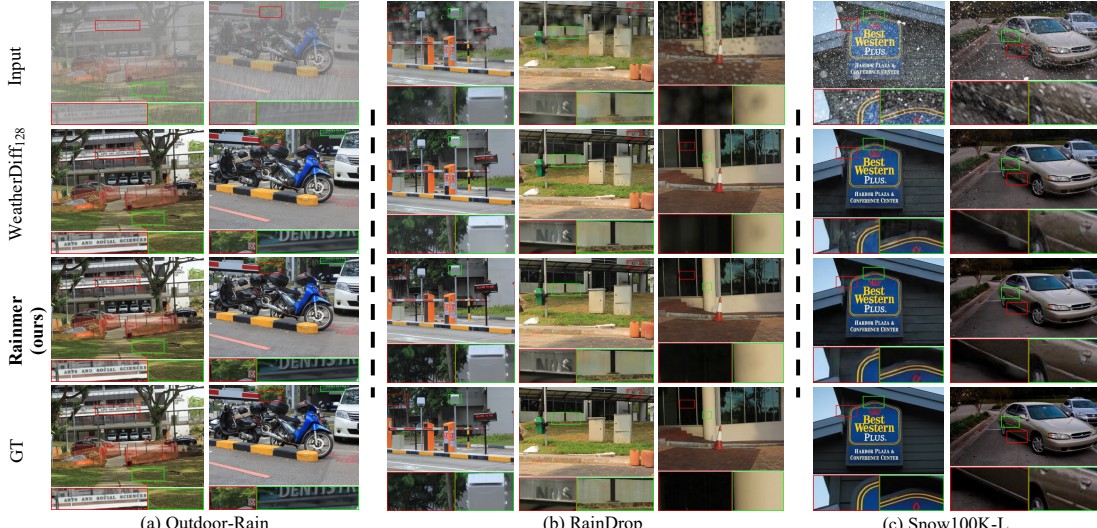

Figure 6: Visual comparison with WeatherDiff$_{128}$ on Outdoor-Rain, RainDrop, and Snow100K-L datasets.

Table 2: All-in-One image restoration comparison. The best and second results are bolded and underlined, respectively.

| Method | Outdoor-Rain | | RainDrop | | Snow100K-L | |
|---|---|---|---|---|---|---|
| | PSNR↑ | SSIM↑ | PSNR↑ | SSIM↑ | PSNR↑ | SSIM↑ |
| NAS [21] (CVPR'20) | 24.71 | 0.8980 | 31.12 | 0.9268 | 28.33 | 0.8820 |
| TransWeather [39] (CVPR'22) | 28.83 | 0.9000 | 30.17 | 0.9157 | 29.31 | 0.8879 |
| WeatherDiff$_{64}$ [31] (TPAMI'23) | 29.64 | 0.9312 | 30.71 | 0.9312 | 30.09 | 0.9041 |
| WeatherDiff$_{128}$ [31] (TPAMI'23) | 29.72 | 0.9216 | 29.66 | 0.9225 | 29.58 | 0.8941 |
| WGWS-Net [67] (CVPR'23) | 30.60 | 0.9646 | **33.26** | **0.9759** | 31.24 | 0.9335 |
| Rainmer (**ours**) | **31.81** | **0.9671** | 32.01 | 0.9679 | **31.50** | **0.9350** |

Table 3: Quantitative comparison of downstream object detection on daytime rainy datasets. We bold the best results and underline the second results.

| Methods | Test100 | | | Test1200 | | |
|---|---|---|---|---|---|---|
| | mAP$_{.50}$ | mAP$_{.75}$ | mAP$_{.50:.95}$ | mAP$_{.50}$ | mAP$_{.75}$ | mAP$_{.50:.95}$ |
| Rainy | 53.87% | 50.03% | 46.98% | 36.28% | 33.80% | 31.19% |
| PReNet [37] | 59.91% | 55.31% | 51.97% | 53.17% | 49.66% | 46.62% |
| BRN [36] | 65.84% | 62.69% | 58.91% | 55.32% | 52.08% | 48.71% |
| RCDNet [43] | 67.89% | 65.07% | 59.40% | 53.41% | 50.45% | 46.67% |
| EfDerain [11] | 69.52% | 63.53% | 61.37% | 56.16% | 52.48% | 48.97% |
| AirNet [19] | 69.67% | 65.75% | 61.55% | 55.23% | 51.77% | 47.93% |
| HCT-FFN [6] | 72.59% | 66.29% | 63.02% | 56.58% | 53.02% | 49.47% |
| DRSformer [4] | 72.50% | 67.53% | 65.30% | 62.38% | **59.65%** | 55.22% |
| PromptIR [33] | 72.90% | 66.62% | 63.71% | 60.27% | 56.86% | 53.09% |
| CoIC [35] | 72.54% | 70.27% | 65.59% | **62.79%** | 59.36% | 55.12% |
| Rainmer (**ours**) | **75.91%** | **70.46%** | **68.82%** | 62.76% | 59.63% | **55.60%** |

surpass previous methods, *e.g.*, CoIC on GT-Rain, which may be attributed to the limitations of constructing negatives by linear addition. Fig. 4 provides a visual comparison on both daytime and nighttime datasets. The results on the first two rows in Fig. 4 indicate that Rainmer can better restore image details. For nighttime image deraining, Rainmer is capable of efficiently removing complex and colored rain streaks. Both the quantitative and qualitative results have substantiated the efficacy of the proposed method.

Fig. 5 provides six real-world deraining examples. Typically, both the unsupervised NLCL and the recent DRSformer struggle to remove complex rain effects compared to the proposed Rainmer. This indicates that Rainmer exhibits strong real-world deraining ability.

### 4.3 Comparison on Image Restoration

We further conduct experiments on the AllWeather dataset to verify the effectiveness of Rainmer. We select NAS [21], TransWeather [39], WeatherDiff$_{64}$ [31], WeatherDiff$_{128}$ [31], and WGWS-Net [67] for comparison. Specifically, TransWeather employs learnable weather queries for prompt image restoration. Table 2 reports quantitative results. The proposed Rainmer has dramatically outperformed all methods on Outdoor-Rain and Snow100k-L datasets, offering 1.21dB and 0.26dB PSNR improvements against the previous best method, respectively. However, Rainmer has not achieved

the best performance on RainDrop, which may be attributed to the imprecise calculation of raindrop layers when synthesizing degradation-aware negatives. We also provide a visual comparison in Fig. 6. Compared to the proposed Rainmer, the recent WeatherDiff$_{128}$ cannot correct background color under rain veiling effect in Outdoor-Rain. Moreover, WeatherDiff$_{128}$ fails to remove raindrop and snow effects in Fig. 6 (b) & (c). These observations assess the superiority of the proposed approach in image restoration.

### 4.4 Improvement on Object Detection

While removing rain effects efficiently, image deraining methods may not consistently improve object detection performance under rainy scenarios [22]. Therefore, we further investigate object detection on daytime datasets (Test100 and Test1200) and nighttime datasets (GTAV-NightRain (set3)). The real-world GT-Rain is excluded due to incomplete content by cropping. Specifically, we utilize the recent RTMDet [28] for detection. Following [18], we

**Table 4: Quantitative comparison of downstream object detection on nighttime rainy dataset. We bold the best results and underline the second results.**

| Methods | GTAV-NightRain (set3) | | |
|---|---|---|---|
| | mAP$_{.50}$ | mAP$_{.75}$ | mAP$_{.50:.95}$ |
| Rainy | 24.09% | 22.51% | 21.48% |
| PReNet [37] | 33.68% | 32.35% | 30.59% |
| BRN [36] | 40.42% | 38.07% | 36.35% |
| RCDNet [43] | 33.79% | 30.83% | 30.17% |
| EfDerain [11] | 34.99% | 32.64% | 31.35% |
| AirNet [19] | 29.79% | 28.32% | 26.54% |
| HCT-FFN [6] | 36.98% | 34.64% | 33.06% |
| DRSformer [4] | 46.46% | 44.52% | 42.55% |
| PromptIR [33] | 44.80% | 41.88% | 40.43% |
| CoIC [35] | 47.65% | 45.88% | 43.54% |
| Rainmer (**ours**) | **51.47%** | **48.69%** | **46.73%** |

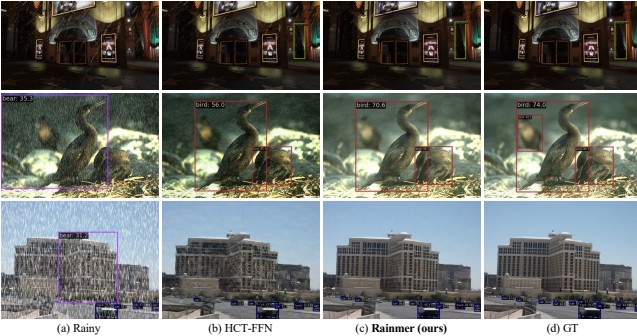

| (a) Rainy | (b) HCT-FFN | (c) **Rainmer (ours)** | (d) GT |

**Figure 7: Object detection examples. The first row: GTAV-NightRain (set3). The last two rows: Test100.**

generate labels from detection results on clean images while excluding low-confidence results. Quantitative results[1][2] on daytime and nighttime datasets are reported in Tables 3 and 4, respectively. Notably, Rainmer has obtained the highest mAP$_{.50:.95}$ metric over all datasets. A visual comparison on nighttime rainy image is presented in Fig. 7, where HCT-FFN fails to detect the "car" object while Rainmer successfully detects all objects. Summarizing results in Tables 1, 3 and 4, we further investigate the correlation between improvements of image quality metrics and mAP metric. The result is presented in Fig. 8, where we observe that higher PSNR and SSIM improvements cannot always bring a higher mAP value. *Only with a significant gain in PSNR/SSIM can we achieve a higher mAP metric.*

### 4.5 Ablation Study

In this section, we conduct experiments to investigate the effectiveness of the proposed multi-view (multiple) representations, SCIM, and prompt weighting strategy. Specifically, we treat Rainmer without multi-view representations, MPM, and SCIM as our baseline.
**Effect of Multi-view Representations**. We first examine the efficacy of multi-view representations. As illustrated in Table 5, Rainmer produces degraded average PSNR metrics when removing

---

[1]IDT [50] is excluded due to high inference burden with fixed input size $128 \times 128$.
[2]unsup. NLCL [56] is excluded due to its bad performance.

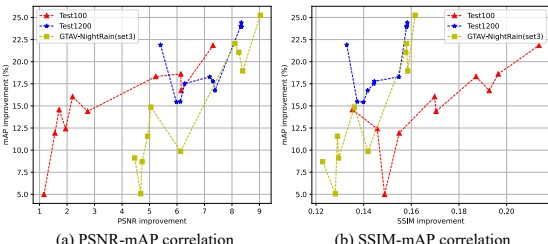

| (a) PSNR-mAP correlation | (b) SSIM-mAP correlation |

**Figure 8: Correlation between PSNR/SSIM and mAP metrics.**

**Table 5: Ablation on multi-view representations and SCIM.**

| Configuration | Rain100L | Rain100H | Test100 | Test1200 | Test2800 | Avg. |
|---|---|---|---|---|---|---|
| Baseline | 35.60 | 29.14 | 29.39 | 31.94 | 33.08 | 31.83 |
| Rainmer w/o $z^{chr}$ | 36.33 | 29.32 | 29.54 | 32.28 | 33.22 | 32.14 |
| Rainmer w/o $z^{det-g}$ | 36.10 | 29.29 | 29.86 | 32.10 | 33.20 | 32.11 |
| Rainmer w/o SCIM | 36.21 | 29.13 | 29.84 | 31.94 | 33.23 | 32.07 |
| Rainmer | 36.42 | 29.38 | 29.86 | 31.98 | 33.22 | **32.17** |

**Table 6: Ablation on prompt weighting in MPM.**

| $\alpha(z^{chr}, z^p)$ | Rain100L | Rain100H | Test100 | Test1200 | Test2800 | Avg. |
|---|---|---|---|---|---|---|
| ✗ | 36.07 | 29.04 | 29.93 | 32.14 | 33.25 | 32.09 |
| ✓ | 36.42 | 29.38 | 29.86 | 31.98 | 33.22 | **32.17** |

one representation. This demonstrates that multi-view representation is necessary to acquire superior performance.
**Effect of SCIM**. As tabulated in Table 5, without SCIM, Rainmer has suffered significant performance drops on Rain100L and Rain100H datasets, resulting in average PSNR degradation. This observation verifies the necessity of SCIM.
**Effect of Prompt Weighting in Eq. (19)**. As shown in Table 6, Rainmer without weighting has undergone severe PSNR metric drops on Rain100L and Rain100H datasets, offering even worse performance over Rainmer only with $z^{chr}$ representation. This observation indicates that directly adding prompting results degrades performance, which can be mitigated with a weighting strategy.

## 5 Conclusion and Future Work

We address more challenging and practical image deraining by leveraging synthetic, real-world, and nighttime datasets. In this scenario, complex backgrounds, diverse rain effects, and varying illuminations coexist, potentially inducing competition and conflicts. To tackle this problem, we propose to learn multi-view (multiple) representations that efficiently perceive image details, degradations, and illuminations. Then, we develop Rainmer, a potent deraining model that integrates multiple representations for prompting restoration. Extensive experiments on synthetic, real-world, and nighttime image deraining, as well as All-in-One image restoration, substantiate the efficacy of Rainmer. Furthermore, Rainmer has been proven to significantly improve object detection performance. In the future, we will develop more effective representation learning methods for universal image restoration under various extreme conditions.

## Acknowledgments

This work was supported by the National Natural Science Foundation of China (No.62072112), the Scientific and Technological Innovation Action Plan of Shanghai Science and Technology Committee (No.22511102202), and the National Key R&D Program of China (2020AAA0108301).

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
