# OpenReview forum: "Rainmer: Learning Multi-view Representations for Comprehensive Image Deraining and Beyond"
_acmmm.org/ACMMM/2024/Conference — MM2024 Poster_

### Official Review · Reviewer_1ADU · 2024-05-16

**Rating:** 3
**Confidence:** 3

**Summary:**

This paper proposes an approach for image deraining under various, complex scenarios, including synthetic, real-world and nighttime data. They devise a framework consisting of data distribution alignment module, a contrastive learning for multi-view representation learning, a spatial-channel interaction module. Experiments on synthetic, real-world and nighttime datsets are conducted to validate its superior performance. Evaluation on downstream tasks is also displayed to illustrate its practical application.

**Strengths:**

It is interesting and practical to design a more generalizable method for adverse weather removal tasks.
A large amount of experiments are conducted on multiple datasets to prove the versatility of the proposed method.
Evaluation on downstream tasks like object detection is great, demonstrating the practical applications of the proposed method.

**Limitations:**

1. This paper proposes an approach for comprehensive image deraining. However, there is no part tailored towards the characteristics of rain streak. Why not directly develop an All-in-One method considering deraining, dehazing and desnowing, similar to NAS, TransWeather? The problem setting is strange.
2. The definition of ``multi-view'' is ambiguous. It is hard to understand why chromatic, detail and degradation share such relation.
3. More All-in-One methods should be compared in Tab. 2, such as TKL, WGWS-Net.
4. Fig. 7 is surprising to me. The performance oscillation happened without regular patterns. More analysis should be given why a better PSNR/SSIM even brings a significant performance drop on downstream tasks.
5. More cases should be given in Fig. 6 to better illustrate practical applications.
6. Visualization results on real-world data are highly desired because this work focuses on generalizable image deraining.
7. Observed from Tab. 5, each proposed module contributes very limited performance gain. It seems the whole framework is designed based on a strong baseline.

**Suitability:**

3

---

### Official Review · Reviewer_D7ti · 2024-05-19

**Rating:** 4
**Confidence:** 4

**Summary:**

This paper proposes a method called Rainmer for comprehensive image deraining. The proposed method aims to address the challenges posed by synthetic, real-world, and nighttime datasets by learning multi-view representations that effectively perceive image details, degradations, and illuminations. Experimental results demonstrate that Rainmer outperforms state-of-the-art methods on various datasets, including synthetic, real-world, and nighttime datasets. The proposed method also achieves a significant improvement in mean Average Precision (mAP) on both daytime and nighttime datasets for object detection tasks.

**Strengths:**

1.	The writing is well and the algorithmic details are clear.
2.	Solving the problem of deraining under various scenes is practical.
3.	The idea of multi-view representation learning is insightful.
4.	The results show the superior performance on each deraining dataset and detection dataset.

**Limitations:**

1.	Why the authors emphasize the U-Net-based design? Is it one of your contributions? It maybe not proper to call a network with attention blocks as U-Net-based architecture.
2.	I can understand that 8 x 8 kernel better preserve the spatial information than global manner, but with an 8 times down-sampling process, the detail information will still be destroyed. And no relative ablation study appears.
3.	Why the authors group detail and degradation representations? I suppose that the details are more related to the content of the image, which should be consistent when meeting different degradations. The reason of this joint training manner is not clear.
4.	In Fig. 4, why only the bottom two rows have zoom-in box?
5.	The effectiveness of prompting weighting is limited.

**Suitability:**

2

---

### Official Review · Reviewer_Pbje · 2024-05-20

**Rating:** 4
**Confidence:** 4

**Summary:**

The authors propose a method that leverages synthetic, real-world, and nighttime datasets to train a more robust deraining model. They address the challenges posed by the discrepancies among these datasets by aligning their distributions.
A contrastive learning strategy is introduced to extract multi-view representations that capture various aspects of the image, including details, degradations, and illuminations.
The paper introduces Rainmer, a U-Net-based deraining model. This model incorporates multi-view representations as prompts during the decoding process to enhance its deraining capability.
Extensive experiments on three datasets demonstrate that Rainmer outperforms existing state-of-the-art methods. The model also shows improvements in object detection performance in both daytime and nighttime rainy conditions.

**Strengths:**

This paper utilize multi-view representations extracted through a  contrastive learning strategy.
By aligning distributions of synthetic, real-world, and nighttime datasets and integrating multi-view representations into the decoding process, Rainmer outperforms existing methods on various datasets, including All-in-One image restoration, and significantly improves object detection performance in both daytime and nighttime rainy scenarios.

**Limitations:**

1. The reviewer notices that Rainmer does not achieve the best results on the GT Rain dataset and there is a significant gap compared to the second method. Can the author discuss the details of the results on this dataset?
2. The Related Work section should be enriched by incorporating relevant literature on contrastive learning in the field, which will enhance the discussion of the innovative aspects of the proposed work. For example：
[1] Ye Y, Yu C, Chang Y, et al. Unsupervised deraining: Where contrastive learning meets self-similarity[C]//Proceedings of the IEEE/CVF conference on computer vision and pattern recognition. 2022: 5821-5830.
[2] Wu H, Qu Y, Lin S, et al. Contrastive learning for compact single image dehazing[C]//Proceedings of the IEEE/CVF conference on computer vision and pattern recognition. 2021: 10551-10560.
3. What does the symbol E_(x, y) in Formula 4 represent?
4. Since this is a new setting for image rain removal, can the author provide comparison results on real images to verify the effectiveness of the method.

**Suitability:**

3

---

### Official Review · Reviewer_e4hM · 2024-05-25

**Rating:** 4
**Confidence:** 3

**Summary:**

The paper presents "Rainmer," a U-Net-based model for image deraining that employs multi-view representations to address complex real-world and nighttime scenarios. It introduces a contrastive learning strategy to capture details, degradations, and illuminations, and integrates these representations into the decoding process with a prompting strategy. Rainmer also includes a Spatial-channel Interaction Module (SCIM) for enhanced feature extraction. Experiments show Rainmer outperforms current methods, with improved object detection capabilities in rainy conditions.

**Strengths:**

1. It captures a comprehensive set of image attributes including details, degradations, and illuminations by learning from diverse datasets, which significantly enhances the model's ability to handle complex deraining tasks.

2. Rainmer demonstrates superior performance over existing methods, with notable improvements in object detection accuracy in rainy conditions, highlighting its practical utility and effectiveness in real-world applications.

**Limitations:**

1. The model might be optimized for heavy rain scenarios and could potentially perform less effectively on images with minor or light rain effects.
2. Unsupervised methods often demonstrate superior performance in deraining natural scenes due to their reduced risk of overfitting to synthetic datasets. Conducting additional comparative experiments with other unsupervised learning models can further validate the model's advantages in natural settings. For instance, comparisons with models such as "Unsupervised Image Deraining Optimization Model Driven Deep CNN",  "UConNet: Unsupervised Controllable Network for Image and Video Deraining" and "Unsupervised deraining: Where contrastive learning meets self-similarity" on datasets featuring natural scenes could be insightful. Assessments could utilize no-reference metrics or alternative evaluation methods to ascertain the model's strengths in real-world scenarios.

**Suitability:**

3

---

### Meta-Review · Area_Chair_RgAu · 2024-07-01

**Recommendation:** Accept (Poster)
**Confidence:** 5

**Metareview:**

This paper was reviewed by four experts in the field. The paper received mixed reviews BA, BA, BA, BR. The reviewers approved the performance of the proposed method. The reviewers also raised several concerns including the significance of the task solved in the paper. Based on the reviews, the AC would like to recommend the acceptance of this paper and suggest the authors include the added experiments in the rebuttal to the final version. However, the authors should add a more detailed discussion about the problem itself and include more all-in-one methods in comparison.